# Improving TD3-BC: Relaxed Policy Constraint for Offline Learning and Stable Online Fine-Tuning

**Alex Beeson**[1]     **Giovanni Montana**[1,2]
[1]University of Warwick     [2]Alan Turing Institute
{alex.beeson,g.montana}@warwick.ac.uk

## Abstract

The ability to discover optimal behaviour from fixed data sets has the potential to transfer the successes of reinforcement learning (RL) to domains where data collection is acutely problematic. In this offline setting, a key challenge is overcoming overestimation bias for actions not present in data which, without the ability to correct for via interaction with the environment, can propagate and compound during training, leading to highly sub-optimal policies. One simple method to reduce this bias is to introduce a policy constraint via behavioural cloning (BC), which encourages agents to pick actions closer to the source data. By finding the right balance between RL and BC such approaches have been shown to be surprisingly effective while requiring minimal changes to the underlying algorithms they are based on. To date this balance has been held constant, but in this work we explore the idea of tipping this balance towards RL following initial training. Using TD3-BC, we demonstrate that by continuing to train a policy offline while reducing the influence of the BC component we can produce refined policies that outperform the original baseline, as well as match or exceed the performance of more complex alternatives. Furthermore, we demonstrate such an approach can be used for stable online fine-tuning, allowing policies to be safely improved during deployment.

## 1   Introduction

In offline reinforcement learning (offline-RL) [15] an agent seeks to discover an optimal policy purely from a fixed data set. This is a challenging task as, unlike traditional reinforcement learning, an agent can no longer interact with the environment to assess the quality of exploratory actions and thus cannot correct for errors in value estimates. These errors can propagate and compound during training leading to highly sub-optimal policies or, in the worst case, a complete failure to learn [16].

A common approach to overcoming this problem is to perform some kind of regularisation during training, encouraging updates during policy evaluation and/or policy improvement to stay close to underlying data. One way to achieve this is to modify existing off-policy algorithms to directly incorporate behavioural cloning (BC), a technique that has been shown to be computationally efficient, easy to implement and able to produce policies that compete with more complex alternatives [6]. Furthermore, such approaches require minimal changes to the underlying algorithms they are based on, making performance gains easier to attribute and hyperparameter tuning less burdensome, both desirable properties when online interaction is limited.

Finding the optimal level of behavioural cloning is key: too little runs the risk of error propagation, but too much limits the agent's ability to learn policies beyond those which generated the data. To date, such approaches have maintained a constant level throughout training, but in this work we explore the idea of reducing the influence of behavioural cloning following an initial period of learning. Using TD3-BC as the foundation, we show that by continuing to train offline in this way we

Offline Reinforcement Learning Workshop at Neural Information Processing Systems, 2022.

are able to produce refined policies that improve over the baseline, particularly in cases where data is sub-optimal.

Deployment of policies following offline training opens up the possibility of further improvement via online fine-tuning. Such a transition should ideally result in negligible performance drops, but the sudden change in setting often means policies get worse before they get better [24]. We investigate whether our offline approach can also be extended to this setting and find we are able to fine-tune both baseline and refined policies while largely mitigating these stability issues.

For both offline learning and online fine-tuning our approach maintains the simplicity of TD3-BC, yet we are able to match or exceed the performance of state-of-the-art alternatives.

## 2   Background

We follow previous work and define a Markov decision process (MDP) with state space $S$, action space $A$, transition dynamics $T(s'|s, a)$, reward function $R(s, a)$ and discount factor $\gamma \in [0, 1]$ [20]. An agent interacts with this MDP by following a policy $\pi(a|s)$, which can be deterministic or stochastic. The goal of reinforcement learning is to discover an optimal policy $\pi^*(a|s)$ that maximises the expected discounted sum of rewards $\mathbb{E}_\pi(\sum_{t=0}^\infty \gamma^t r(s_t, a_t))$.

In Actor-Critic methods this is achieved by alternating between policy evaluation and policy improvement using Q-functions $Q^\pi(s, a)$, which estimate the value of taking action $a$ in state $s$ following policy $\pi$ thereafter. Policy evaluation consists of updating Q-values based on the Bellman expectation equation $Q^\pi(s, a) = r(s, a) + \gamma \mathbb{E}_{s' \sim T, a' \sim \pi}(Q^\pi(s', a'))$ with improvement coming from updating policy parameters to maximise $Q^\pi(s, a)$.

In offline reinforcement learning the agent no longer has access to the environment and instead must learn solely from pre-existing interactions $D = (s_i, a_i, r_i, s_i')$. In this setting, agents struggle to learn effective policies due to overestimation bias for state-action pairs absent in $D$, which compounds and propagates during training leading to highly sub-optimal policies or a complete collapse of the learning process [16].

TD3-BC [6] is an adaptation of TD3 [7] to the offline setting, adding a behavioural cloning term to policy updates to encourage agents to stay close to actions in $D$. The updated policy objective is:

$$\pi = \arg\max_\pi \mathbb{E}_{(s,a) \sim D} \Big[ \ \underbrace{Q(s, \pi(s))}_{\text{reinforcement learning}} \ - \ \alpha \ \underbrace{(\pi(s) - a)^2}_{\text{behavioural cloning}} \ \Big].$$

The hyperparamter $\alpha$ controls the balance between reinforcement learning and behavioural cloning. In order to keep this balance in check, Q-values (which scale with rewards and hence can vary across tasks) are normalised by dividing by the mean of the absolute values. States are also normalised to have mean 0 and standard deviation 1:

$$Q_{norm}(s_i, a_i) = \frac{Q(s_i, a_i)}{\frac{1}{N} \sum_{s_i, a_i} |Q(s_i, a_i)|} \qquad , \qquad s_i = \frac{s_i - \mu_i}{\sigma_i + \epsilon},$$

where $\epsilon = 10^{-3}$ is a small normalisation constant.

The only remaining task it to find a value of $\alpha$ that both prevents the propagation of overestimation bias and allows the agent to learn a policy that improves on the data.

## 3   Related work

While TD3-BC uses behavioural cloning explicitly in policy updates, other approaches such as BCQ [8], PLAS [22], BRAC [22] and BEAR [13] utilise it more implicitly, training a Conditional Variational AutoEncoder (CVAE) [19] to mimic the data distribution and either optimise CVAE generated actions (BCQ, PLAS) or minimise divergence metrics between the CVAE and the policy (BRAC, BEAR). Such methods do successfully combat overestimation bias, but tend to perform worse than simpler methods such as TD3-BC, particularly for multimodal distributions which are hard to model accurately.

Other examples of using behavioural cloning less directly include Implicit Q-learning (IQL) [12] which combines expectile regression and advantage weighted behavioural cloning to train agents without having to evaluate actions outside the dataset, and Fisher-BRC [11] which uses a cloned behaviour policy for critic regularisation via the Fisher divergence metric. Both methods perform comparably to TD3-BC but also suffer to the same degree when learning from sub-optimal data.

Conservative Q-learning (CQL) [14] focuses specifically on critic regularisation by augmenting the Bellman expectation equation by "pushing down" on $Q(s, a)$ for actions outside the dataset and "pushing up" on $Q(s, a)$ for actions in the dataset. CQL performs well across a wide range of tasks but is computationally expensive and often relies on implementation tweaks to achieve such levels of performance [6].

Online fine-tuning following offline learning has largely focused on final performance, but approaches such as REDQ+AdaptiveBC [24] also recognise the importance of stability. By adapting the BC component based on online returns, REDQ+AdaptiveBC is able to improve policies online in a reasonably stable manner, although requires the use of ensembles and domain knowledge to do so.

Finally, we note that behavioural cloning has also been used in the online setting with a focus on data efficiency, such as combining with DDPG [18] or SAC [17].

## 4 TD3-BC improvements

### 4.1 Offline policy refinement

The relative simplicity and competitive performance of TD3-BC makes it an attractive proposition for offline-RL. There are however occasions where it falls short, particularly when the underlying data is sub-optimal. The BC component which is so crucial for preventing overestimation bias is the same component which limits the agent's potential. *Is there a way to overcome this limitation without increasing the algorithm's complexity?*

We note that, at the start of training, too low a value of $\alpha$ fails to prevent overestimation bias. When $\alpha$ is sufficiently high, this overestimation bias is mitigated leading to a more accurate critic, albeit limited to the state-action region contained in $D$. However, in learning to produce reasonable estimates in this region, it may then be possible for the critic to improve its estimates outside this range, allowing the value of $\alpha$ to be reduced to a level which at the start was too small. In other words, the critic learns from actions close to the ground truth before generalising to actions further away: learning to walk before learning to run.

With this in mind we modify the training procedure of TD3-BC to include an additional period of policy refinement, where we continue to train offline with a reduced value of $\alpha$. To prevent overestimation bias reappearing we focus solely on the policy, performing no further updates to the critic. We illustrate this procedure in Figure 1 and provide full details in Section 4.3.

While there are many options available for reducing $\alpha$, for the purpose of this work we simply scale by a constant $\lambda \geq 1$, giving rise to a new BC coefficient $\alpha'$ such that:

$$\alpha' = \frac{\alpha}{\lambda}.$$

(1)

### 4.2 Online fine-tuning

Once deployed, an agent has the opportunity for continued improvement via online fine-tuning. While one option is to simply remove the BC constraint imposed during offline learning, this often leads to an initial period of policy degradation owing to the sudden shift from the offline to online setting. In many situation this would be deemed unacceptable, hence approaches that promote stability as well as performance are desired.

In light of this, we devise an alternative approach for transitioning to the online setting. We initialise a new replay buffer and train both critic and policy on additional interactions collected from the environment. The influence of the BC term is reduced based on exponential decay with decay rate $\kappa_\alpha$:

$$\kappa_\alpha = \exp\left[\frac{1}{N}\log\left(\frac{\alpha_{end}}{\alpha_{start}}\right)\right].$$

(2)

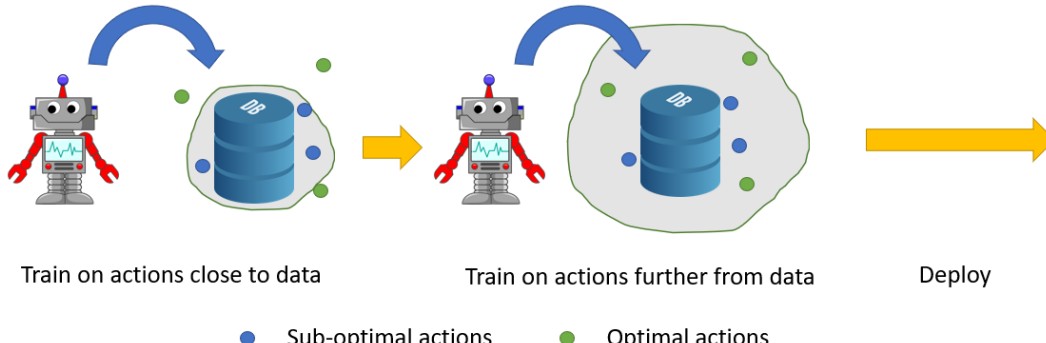

Train on actions close to data     Train on actions further from data     Deploy

● Sub-optimal actions     ● Optimal actions

Figure 1: Policy refinement. To begin with, an agent is trained on actions close to the data to avoid overestimation bias. Next, the agent is trained on actions further from the data to refine the policy prior to deployment. Following this procedure allows better generalisation to unseen state-action pairs that an agent may poorly estimate without prior training. Grey area represents action space permitted by the value of $\alpha$, with blue and green dots representing sub-optimal and optimal actions, respectively.

Here, $\alpha_{start}$ and $\alpha_{end}$ are the initial and final values of $\alpha$, respectively, and $N$ is the number of decay steps. Full details are again provided in Appendix 4.3.

We adopt this approach over REDQ+AdaptiveBC to avoid the need for domain knowledge and ensemble methods, staying as close as possible to the original TD3-BC algorithm.

### 4.3 Algorithms

For completeness, in Algorithms 1 and 2 we outline our policy refinement and online fine-tuning procedures. Corresponding parameter updates and notation are as follows.

Let $\theta_i$ and $\theta_i'$ represent the parameters of the $i^{th}$ Q-network and target Q-network, respectively, and $\phi$ and $\phi'$ represent the parameters for a policy network and target policy network, respectively. Let $\alpha$ represent the BC coefficient, $\tau$ the target network update rate, $\epsilon$ policy noise variance and $B$ a sample of transitions from dataset $D$.

Each Q-network update is performed through gradient descent using:

$$\nabla_{\theta_i} \frac{1}{|B|} \sum_{(s,a,r,s') \in B} \left( Q_{\theta_i} - r - \gamma \min_{i=1,2} Q_{\theta_i}(s', a') \right)^2, \tag{3}$$

where $a' = \text{clip}(\pi_{\phi'}(s') + \text{noise}, \text{-}0.5, 0.5)$ with noise sampled from an $N(0, \epsilon)$ distribution.

The policy network update is performed through gradient ascent using:

$$\nabla_{\phi} \frac{1}{|B|} \sum_{(s,a) \in B} Q_{\theta_1}\left( s, \pi_{\phi}(s) \right) - \alpha \left( \pi_{\phi}(s) - a \right)^2. \tag{4}$$

Finally, target networks are updated using Polyak averaging:

$$\begin{aligned} \theta_i' &\leftarrow \tau \theta_i + (1 - \tau)\theta_i' \\ \phi' &\leftarrow \tau \phi + (1 - \tau)\phi'. \end{aligned} \tag{5}$$

---
**Algorithm 1** TD3-BC with policy refinement
---
**Require:** Dataset $D$, discount factor $\gamma$, target update rate $\tau$, policy noise variance $\epsilon$ and BC coefficient $\alpha$

Initialise critic parameters $\theta_i$, policy parameters $\phi$ and corresponding target parameters $\theta_i'$, $\phi'$.

**TD3-BC**
**for** $j = 0$ to $J$ **do**
    Sample minibatch of transitions $(s, a, r, s')$ from $D$
    Update critic parameters $\theta_i$ using equation (3)
    Update policy parameters $\phi$ using equation (4)
    Update target network parameters $\theta_i'$, $\phi'$ using equations (5)
**end for**

**Policy refinement**
Set scaling factor $\lambda$ and replace $\alpha$ with $\alpha'$ as per equation 1
**for** $k = 0$ to $K$ **do**
    Sample minibatch of transitions $(s, a, r, s')$ from $D$
    Update policy parameters $\phi$ using equation (4)
    Update target network parameters $\phi'$ using equations (5)
**end for**
---

---
**Algorithm 2** Online fine-tuning
---
**Require:** Discount factor $\gamma$, target update rate $\tau$, exploration noise $\sigma$, decay parameters $\alpha_{start}, \alpha_{end}, N$ and BC coefficient $\alpha$

Initialise pre-trained critic parameters $\theta_i$, policy parameters $\phi$ and corresponding target parameters $\theta_i'$, $\phi'$.

Initialise environment and replay buffer $R$.
Set decay rate $\kappa_\alpha$ as per equation 2
Set $\alpha = \alpha_{start}$
**for** $m = 0$ to $M$ **do**
    Act in environment with exploration, $a \sim \pi_\phi(s) + N(0, \sigma)$
    Store resulting transition $(s, a, r, s')$ in $R$
**end for**
**for** $n = 0$ to $N$ **do**
    Act in environment with exploration, $a \sim \pi_\phi(s) + N(0, \sigma)$
    Store resulting transition $(s, a, r, s')$ in $R$
    Sample minibatch of transitions $(s, a, r, s')$ from $R$
    Update critic parameters $\theta_i$ using equation (3)
    Update policy parameters $\phi$ using equation (4)
    Update target network parameters $\theta_i'$, $\phi'$ using equations (5)
    Update decay rate $\alpha \leftarrow \kappa_\alpha \alpha$
**end for**
---

## 5   Experiments

We evaluate our method using MuJoCo tasks from the open-source D4RL[1] benchmark [21] [5] and compare to a number of state-of-the-art approaches mentioned in Section 3. This benchmark contains a variety of tasks aimed at testing an agent's ability to learn from sub-optimal or narrow data distributions as well as expert demonstrations. For the tasks considered in this paper the data sets comprise the following

- **Expert** 1M interactions collected from an agent trained to expert level using SAC
- **Medium** 1M interactions collected from an agent trained to 1/3 expert level using SAC
- **Medium-Expert** The combined interactions from expert and medium (2M total)
- **Medium-Replay** The replay buffer from training the medium-level agent (variable size)

---

[1]https://github.com/Farama-Foundation/D4RL

Each experiment is repeated across five random seeds and we report the mean normalised score $\pm$ one standard deviation for 50 evaluations (10 from each seed). Scores of 0 and 100 represent equivalent random and expert policies, respectively. Further details, including network architecture and hyperparameters, can be found in Appendix A.1.

**Offline policy refinement:** Following the procedure in [6], we train an agent for 1M gradient steps with $\alpha = 0.4$ to produce a baseline policy. We then perform policy refinement for a further 250k steps, setting $\lambda = 5$ for all tasks except hopper-medium-expert and halfcheetah-expert/medium-expert where $\lambda = 1$.

We also conduct a number of ablation studies to highlight the importance of the design decisions outlined in Section 4.1. These results are presented in Appendix A.2.

**Online fine-tuning:** We assess each agent's ability to transition from offline to online learning, focusing on stability as well as performance. We initialise a new replay buffer and populate with 5k $(s, a, r, s')$ samples collected using the offline trained policy. We then train both critic and policy for an additional $N=245$k environment interactions (250k in total), decaying $\alpha$ exponentially from $\alpha_{start} = 0.4$ to $\alpha_{end} = 0.2$.

Results are summarised in Table 1. We denote the baseline policy as TD3-BC and refined policy as TD3-BC-PR. Corresponding online fine-tuned policies are suffixed -FT. Offline results for BC, CQL and IQL are taken from [12] with additional comparisons provided in Appendix A.3.

| Dataset | BC | CQL | IQL | TD3-BC | TD3-BC PR | TD3-BC FT | TD3-BC PR-FT |
|---|---|---|---|---|---|---|---|
| halfcheetah-med | 42.6 | 44.0 | 47.4 | 48.5 $\pm$0.7 | **55.3** $\pm$0.8 | 74.5 $\pm$2.1 | 74.4 $\pm$2.4 |
| hopper-med | 52.9 | 58.5 | 66.3 | 56.6 $\pm$9.4 | **100.1** $\pm$2.8 | 93.7 $\pm$20.6 | 102.8 $\pm$0.9 |
| walker2d-med | 75.3 | 72.5 | 78.3 | 83.3 $\pm$7.0 | 89.1 $\pm$1.7 | 105.5 $\pm$2.6 | 101.2 $\pm$3.6 |
| halfcheetah-med-rep | 36.6 | 45.5 | 44.2 | 44.5 $\pm$0.8 | **48.7** $\pm$1.2 | 68.0 $\pm$2.6 | 69.8 $\pm$1.7 |
| hopper-med-rep | 18.1 | 95.0 | 94.7 | 55.2 $\pm$24.6 | **100.5** $\pm$1.2 | 102.8 $\pm$1.8 | 103.4 $\pm$1.7 |
| walker2d-med-rep | 26.0 | 77.2 | 73.9 | 82.5 $\pm$13.6 | 87.9 $\pm$2.8 | 106.6 $\pm$2.9 | 103.9 $\pm$8.1 |
| halfcheetah-med-exp | 55.2 | 91.6 | 86.7 | 91.5 $\pm$15.8 | 91.9 $\pm$11.1 | 96.2 $\pm$1.3 | 96.6 $\pm$1.1 |
| hopper-med-exp | 52.5 | 105.4 | 91.5 | 101.6 $\pm$23.2 | 103.9 $\pm$15.9 | 110.8 $\pm$6.4 | 111.2 $\pm$5.6 |
| walker2d-med-exp | 107.5 | 108.8 | 109.6 | 110.4 $\pm$0.4 | **112.7** $\pm$0.3 | 117.8 $\pm$1.2 | 119.1 $\pm$2.8 |
| halfcheetah-exp | - | - | - | 97.1 $\pm$1.4 | 97.5 $\pm$1.1 | 96.7 $\pm$2.1 | 97.7 $\pm$1.8 |
| hopper-exp | - | - | - | 112 $\pm$1.0 | 112.4 $\pm$0.8 | 111.8 $\pm$0.8 | 111.9 $\pm$0.7 |
| walker2d-exp | - | - | - | 110.2 $\pm$0.2 | **113.0** $\pm$0.5 | 120.0 $\pm$1.7 | 121.4 $\pm$1.9 |

Table 1: Comparison of results for D4RL benchmark using -v2 data sets. Offline learning: refined policies (PR) match or exceed the performance of TD3-BC baseline and competing methods, particularly for medium/medium-replay tasks. Online learning: Fine-tuning policies (FT) continues to improve performance. Figures are mean normalised scores $\pm$ one standard deviation across 50 evaluations (10 evaluations across 5 random seeds). Bold indicates highest mean for offline learning outside error range of TD3-BC.

For offline learning, we observe that policy refinement produces policies that outperform the original baseline. This is particularly the case when data is sub-optimal, i.e. medium/medium-replay or where TD3-BC performs poorly, i.e. hopper-medium/medium-replay. For data sets where TD3-BC already achieves expert or near-expert performance, refined policies match or exceed this. TD3-BC is now able to compete with or surpass other leading methods across all tasks while remaining the simplest to implement.

For online learning, we see from Table 1 that both baseline and refined policies benefit from fine-tuning in terms of performance. To assess stability, we plot the corresponding learning curves in Figure 2 using un-normalised scores as the performance metric, with evaluation taking place every 5k iterations. The solid line represents the mean, shaded area $\pm$ one standard deviation and dashed line performance prior to fine tuning. Learning is relatively stable with very few severe performance drops and comparable to results presented in [24]. The only exception to this is hopper-medium-expert, which takes nearly the entire training period to stabilise. Note this procedure works equally well for baseline and refined policies, showing there is no detrimental impact of offline policy refinement prior to online fine-tuning.

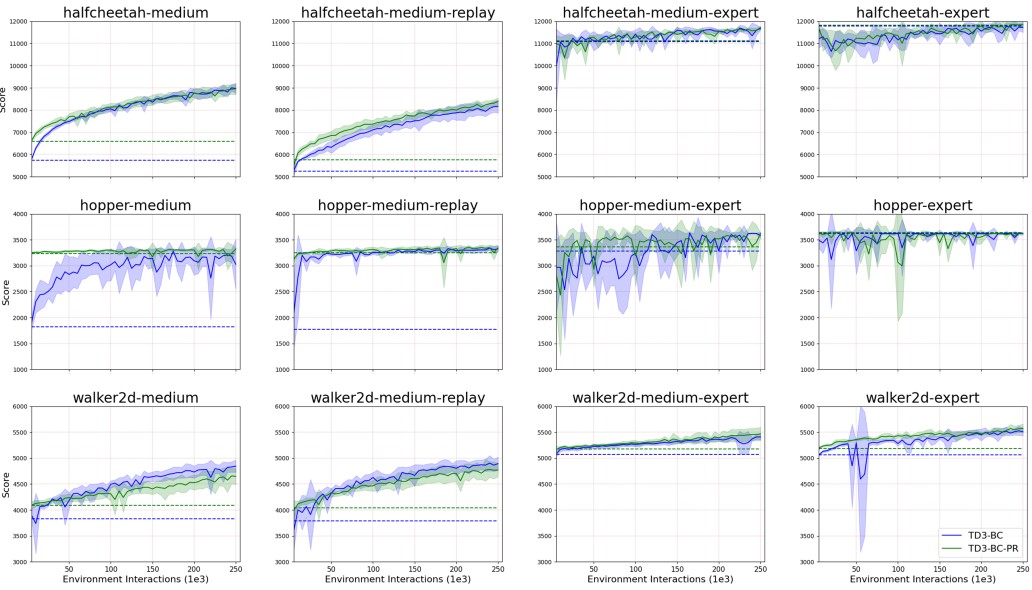

Figure 2: Learning curves for online fine-tuning. For all but hopper-medium-expert fine-tuning is reasonably stable with very few sudden performance drops. This is the case for both baseline policies (blue) and refined policies (green).

# 6   Discussion and conclusion

In this paper we have presented approaches for improving the performance of TD3-BC during offline learning and online fine-tuning. We have shown empirically how deficiencies with TD3-BC can be overcome using a policy refinement procedure and how such an approach can also be used for stable online fine-tuning. In both cases our results match or exceed the performance of state-of-the-art alternatives while being much simpler.

There are however several limitations to our work. First, our policy refinement process is sensitive to the value of $\lambda$, particularly for the medium-expert data sets (see Appendix A.4). While much published research in offline-RL report results for optimised hyperparameters across tasks, such a procedure is unrealistic for real-world application where online fine-tuning is limited or prohibited. Second, our fine-tuning procedure is not completely immune to performance drops, which may be considered unacceptable for high-risk environments. Finally, we have only applied our method to dense-reward environments, but equally important are settings where rewards are sparse.

In future work we look to address these shortcomings as well as conduct a theoretical analysis to complement our empirical results. In addition, we plan to incorporate our approach into online reinforcement learning as a means of improving data efficiency. By showing it is possible to train performant policies through simple modifications to TD3-BC we hope to highlight the potential of minimalist approaches and encourage researchers to explore the possibilities within this framework.

# Acknowledgements

AB acknowledges support from a University of Warwick and University Hospitals Birmingham NHS Foundation Trust collaborative doctoral training scholarship. GM acknowledges support from a UKRI AI Turing Acceleration Fellowship (EPSRC EP/V024868/1). We also acknowledge Weights & Biases [2] as the online platform used for experiment tracking and visualizations to develop insights for this work.

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

# A  Appendix

## A.1  Experimental details

**TD3-BC hyperparameters and network architecture**
In the original TD3-BC [6] the hyperparamater $\alpha$ is attached to Q-values but since our focus in the BC component we attach to the MSE. This is an equivalent set-up if $\alpha = 0.4$, the inverse of the original value $\alpha = 2.5$. All other details are the same, as per Table 2

|  | Hyperparameter | Value |
|---|---|---|
| | Optimizer | Adam |
| | Actor learning rate | 3e-4 |
| | Critic learning rate | 3e-4 |
| | Batch size | 256 |
| | Discount factor $\gamma$ | 0.99 |
| TD3-BC | Target network update rate $\tau$ | 0.005 |
| | Policy noise $\epsilon$ | 0.2 |
| | Policy noise clipping | (-0.5, 0.5) |
| | Critic-to-Actor update ratio | 2:1 |
| | BC regulariser $\alpha$ | 0.4 |
| | Exploration noise $\sigma$ | 0.1 |
| TD3-BC online | $\alpha_{start}$ | 0.4 |
| | $\alpha_{end}$ | 0.2 |
| | Critic hidden nodes | 256 |
| | Critic hidden layers | 2 |
| | Critic hidden activation | ReLU |
| | Critic input | State + Action |
| | Critic output | Q-value |
| Architecture | Actor hidden nodes | 256 |
| | Actor hidden layers | 2 |
| | Actor hidden activation | ReLU |
| | Actor input | State |
| | Actor outputs | Action (tanh transformed) |

Table 2: TD3-BC hyperparameters and network architecture

**Policy refinement hyperparameter tuning**

Table 3 reiterates the value of the scaling parameter $\lambda$ used in our experiments. Our procedure for tuning $\lambda$ emulates that of the TD3-BC paper. We used hopper-medium and hopper-expert as test cases and tuned over range $\lambda = (1, 2, 3, 4, 5)$. Based on the resulting policy performances, we chose $\lambda = 5$ and applied this to the remaining data sets. Performance was improved over the baseline for all data sets apart from hopper-medium-expert, halfcheetah-medium-expert and halfcheetah-expert where performance was significantly worse. For these three, we hypothesised the critic was unable to generalised to actions beyond the region it was trained on and hence reduced the value of $\lambda$ as per the above range. We found that for $\lambda > 1$ policies were still inferior to their baselines and hence $\lambda = 1$ was chosen.

**Offline policy evaluation procedure**
Policy evaluation takes place online in the simulated MuJoCo environment. The reported results are mean normalised scores $\pm$ one standard deviation across evaluations and seeds, where scores of 0 and 100 represent random and expert policies, respectively. Further details can be found in the D4RL paper [5].

**Online policy evaluation procedure**
Policy evaluation takes place online in the simulated MuJoCo environment. The reported results are mean un-normalised scores $\pm$ one standard deviation across seeds

| Dataset | Scaling factor $\lambda$ |
|---|---|
| halfcheetah-medium | 5 |
| hopper-medium | 5 |
| walker2d-medium | 5 |
| halfcheetah-medium-replay | 5 |
| hopper-medium-replay | 5 |
| walker2d-medium-replay | 5 |
| halfcheetah-medium-expert | 1 |
| hopper-medium-expert | 1 |
| walker2d-medium-expert | 5 |
| halfcheetah-expert | 1 |
| hopper-expert | 5 |
| walker2d-expert | 5 |

Table 3: Policy refinement scaling factor $\lambda$

**Hardware** All experiments were conducted on a single machine with the following hardware specification: Intel Core i9 9900K CPU, 64GB RAM, 1TB SSD, 2x NVIDIA GeForce RTX 2080Ti 11GB TURBO GPU.

## A.2   Ablations

In Section 4.1 we note two key designs decisions as part of policy refinement. The first is reducing the value of $\alpha$ after initial training, allowing us to choose values that at the start would be too low. The second is only training the policy while keep the critic fixed in efforts to prevent overestimation bias reappearing. To check the relative importance of each of these we carry out the following ablations:

- Ablation 1: Run baseline TD3-BC with the value of $\alpha'$ from Section 5
- Ablation 2: During policy refinement train both the policy and critic. Note, to reach an equivalent 250k policy updates the total additional gradient steps is 500k as the critic-actor update ratio is 2:1.

For completeness we also run the baseline TD3-BC with $\alpha = 0.4$ for 1.5M gradient steps (1.5M critic, 750k policy) to show additional training alone isn't the cause of policy improvements. We denote this Ablation 3.

Results are summarised in Table 4. Ablation 1 results show that while setting $\alpha = \alpha'$ works for a few data sets, for most this leads to a collapse in learning and highly sub-optimal policies. Ablation 2 results show that training the policy and critic leads to collapse for certain data sets, and even when it doesn't the results aren't as strong as training the policy alone. Ablation 3 results show additional training alone can't match the performance of policy refinement when data sets are sub-optimal (medium/medium-replay).

## A.3   Additional comparisons for offline-RL

In Table 5 we provide additional comparisons for a range of offline approaches, including some model-based methods not covered in Section 3. Specifically we compare to:

- Advantage weighted actor-critic (AWAC) [17]
- One-step Exp. Weight (OnestepRL) [3]
- Decision Transformer (DT) [4]
- Diffuser [9]
- CQL + Model Based Policy Optimization (COMBO) [23]
- Model-based Offline Reinforcement Learning (MOReL) [10]
- Model-based Offline Planning (MBOP) [1]

| Dataset | TD3-BC Refined | Ablation 1 | Ablation 2 | Ablation 3 |
|---|---|---|---|---|
| halfcheetah-med | 55.3 ±0.8 | 55.9 ±1.0 | 55.7 ±1.2 | 48.2 ±0.7 |
| hopper-med | 100.1 ±2.8 | 14.8 ±9.8 | 100.8 ±0.6 | 62.5 ±13.4 |
| walker2d-med | 89.1 ±1.7 | 5.0 ±7.9 | 48.8 ±38.6 | 82.9 ±10.1 |
| halfcheetah-med-rep | 48.7 ±1.2 | 50.2 ±1.1 | 49.4 ±1.0 | 44.6 ±0.7 |
| hopper-med-rep | 100.5 ±1.2 | 90.6 ±9.4 | 88.0 ±23.6 | 55.9 ±25.2 |
| walker2d-med-rep | 87.9 ±2.8 | 41.9 ±34.6 | 70.6 ±35.1 | 77.1 ±21.3 |
| halfcheetah-med-exp | 91.9 ±11.1 | 91.5 ±15.8* | 95.5 ±2.2** | 95.5 ±2.2 |
| hopper-med-exp | 103.9 ±15.9 | 101.6 ±23.2* | 91.6 ±27.8** | 91.6 ±27.8 |
| walker2d-med-exp | 112.7 ±0.3 | 22.6 ±44.4 | 3.9 ±4.7 | 109.8 ±0.9 |
| halfcheetah-exp | 97.5 ±1.1 | 97.1 ±1.4* | 97.6 ±1.4** | 97.6 ±1.4 |
| hopper-exp | 112.4 ±0.8 | 2.9 ±2.3 | 111.1 ±8.5 | 111.7 ±0.7 |
| walker2d-exp | 113.0 ±0.5 | 4.5 ±5.1 | 112.7 ±0.3 | 110.5 ±0.1 |

Table 4: Ablation studies investigating the importance of design decisions in policy refinement. While some data sets are agnostic to these choices, in general they are required to attain the performance levels reported in the paper. The results marked with a single asterisk are the same as baseline TD3-BC and the results marked with a double asterisk are the same as Ablation 3 (as $\lambda = 1$ for these data sets).

We note some comparisons aren't directly equivalent due to differences in data sets, using -v0 instead of -v2. Since we use the latest -v2 we mark -v0 approaches with an asterisk. AWAC, OnestepRL and DT are taken from [12] with Diffuser, COMBO, MOReL and MBOP taken from their respective papers. As can be seen, TD3-BC-PR remains competitive while adhering to the minimalist approach.

| Dataset | TD3-BC-PR | AWAC | OnestepRL | DT | Diffuser | COMBO* | MOReL* | MBOP* |
|---|---|---|---|---|---|---|---|---|
| halfcheetah-med | 55.3 ±0.8 | 43.5 | 48.4 | 42.6 | 44.2 | 54.2 | 42.1 | 44.6 |
| hopper-med | 100.1 ±2.8 | 57.0 | 59.6 | 67.6 | 58.5 | 97.2 | 95.4 | 48.8 |
| walker2d-med | 89.1 ±1.7 | 72.4 | 81.8 | 74.0 | 79.7 | 81.9 | 77.8 | 41.0 |
| halfcheetah-med-rep | 48.7 ±1.2 | 40.5 | 38.1 | 36.6 | 42.2 | 55.1 | 40.2 | 42.3 |
| hopper-med-rep | 100.5 ±1.2 | 37.2 | 97.5 | 82.7 | 96.8 | 89.5 | 93.6 | 12.4 |
| walker2d-med-rep | 87.9 ±2.8 | 27.0 | 49.5 | 66.6 | 61.2 | 56.0 | 49.8 | 9.7 |
| halfcheetah-med-exp | 91.9 ±11.1 | 42.8 | 93.4 | 86.8 | 79.8 | 90.0 | 53.3 | 105.9 |
| hopper-med-exp | 103.9 ±15.9 | 55.8 | 103.3 | 107.6 | 107.2 | 111.1 | 108.7 | 55.1 |
| walker2d-med-exp | 112.7 ±0.3 | 74.5 | 113.0 | 108.1 | 108.4 | 103.3 | 95.6 | 70.2 |

Table 5: Additional comparisons for offline D4RL tasks. Even though many of these methods are more complex and computationally expensive, TD3-BC-PR remains competitive across all tasks. Note results for -expert data sets are not reported for these alternative methods

## A.4 Medium-Expert performance for $\lambda = 5$

As mentioned in Section 6 our policy refinement procedure is sensitive to the quality of the data set. This is particularly the case for hopper/halfcheetah-medium-expert where, as can be seen in Table 6, using $\lambda = 5$ leads to inferior performance than the TD3-BC baseline:

| Dataset | TD3-BC-PR |
|---|---|
| halfcheetah-medium-expert | 63.0 ±27.1 |
| hopper-medium-expert | 69.3 ±39.7 |

Table 6: Policy refinement for hopper and halfcheetah medium-expert dataset when $\lambda = 5$. Performance is significantly worse than TD3-BC baseline

