# OpenReview forum: "Improving TD3-BC: Relaxed Policy Constraint for Offline Learning and Stable Online Fine-Tuning"
_NeurIPS.cc/2022/Workshop/Offline_RL — Offline RL Workshop NeurIPS 2022_

### Official Review · Reviewer_FGdB · 2022-10-17
**Solid idea & core results, paper's writing and presentation can be improved**

**Rating:** 7
**Confidence:** 4

**Review:**

This seems like a good review paper. The central idea is well motivated in the literature and seems to have promising results. There are definitely some things that can be improved in the experiments and presentation, but I will recommend accepting to the workshop.

Strengths:
- the paper is well written conceptually and has a reasonable set of experiments, which can be hard to do in the short workshop format
- the core enhancement of ORL algorithms is simple and shows good results, which is excellent
- the paper has good evaluation practices over random seeds, and I like the scoring between 0 and 100 for random <-> expert

Weaknesses:
- I’m not sure about the title. The phrasing “Carry Offline” is not clearly matched to the central idea of the paper (changing the behavior cloning weight), which makes the paper a lot more confusing to pick up.
- Can figure 1 be iterated on to indicate the intuitions of bias / what is happening to performance too? It’s not immediately clear what is happening (while the alpha shading is interesting, it is important and is not very apparent)
- the fine tuning experiments in the paper do not show clear results. I think they complicate the paper. I understand the motivation for it, but the authors seemed to think their method would help with this and it doesn’t seem to have a notable impact.

Additional Questions:
- can the authors speak further to what tuning was done in the one environment where lambda was =1 (cheetah expert / medium-expert). Conceptually, what was happening here? It is also telling that this is the one environment where the method was not the best. Were all methods tuned extensively?
- What is the implication of adapting BC weights in ORL? How will this change the field in the future? I would like to see the conclusion section be a little more inspiring :)
- What other analyses can be done to showcase the effectiveness of the method? It would be really interesting to me to see how changing the BC weight changes value estimation or something across the state-space.
- Did the authors consider more complex schedules of the alpha parameter instead of just two phases — much like learning rates or diffusion models?


Small issues / nits:
* There are a series of missing punctuations in this paper. I have copied the text for the authors to fix them, most of them need 1+ more comma’s:
“not present in data which, without”
“RL and BC such”
“constant but in this work we explore “
“propagation but”
“combat overestimation bias but tend”
“We note that at the start of training too low “
“When α is sufficiently high this overestimation bias is mitigated leading to a more accurate critic, albeit more”
“produce reasonable estimates in this region it”
* starting a sentence with “But” makes it incomplete
* missing period “λ = 1”
* the authors say “despite the simplicity of our approach” where I would say this is a strength :) — simplicity is good!
* I tend to like to bold numbers in tables when the result is outside of the error band of any other number in the table. It makes the results seem more genuine.
* Figure 2 is very hard to read (too small)

---

### Official Review · Reviewer_kFcR · 2022-10-20

**Rating:** 6
**Confidence:** 3

**Review:**

Summary: TD3-BC (Fujimoto et al., 2021) is a simple offline RL algorithm that balances between TD learning and Behavioral Cloning (BC) to ensure the learned policy distribution stays close to the dataset distribution. The paper builds on TD3-BC and proposes to tip the balance towards TD learning as the training progresses. The resulting method, called TD3-BC PR, improves the performance of offline RL policies on D4RL environments. The paper also claims that the policies trained by TD3-BC PR is better at online finetuning.

Pros: With a small change, the paper improves the performance of TD3-BC across D4RL mujoco environments.

Cons: Given the results, the performance of TD-BC PR after online finetuning looks similar to that of TD3-BC after finetuning. Hence, I am unsure if TD3-BC PR helps in learning policies better at online finetuning.